# Learning to Share in Multi-Agent Reinforcement Learning

**Yuxuan Yi**[1]     **Ge Li**[1]     **Yaowei Wang**[2]     **Zongqing Lu**[1†]
[1]Peking University
[2]Peng Cheng Lab
{touma,geli}@pku.edu.cn     wangyw@pcl.ac.cn     zongqing.lu@pku.edu.cn

## Abstract

In this paper, we study the problem of networked multi-agent reinforcement learning (MARL), where a number of agents are deployed as a partially connected network and each interacts only with nearby agents. Networked MARL requires all agents to make decisions in a decentralized manner to optimize a global objective with restricted communication between neighbors over the network. Inspired by the fact that *sharing* plays a key role in human's learning of cooperation, we propose LToS, a hierarchically decentralized MARL framework that enables agents to learn to dynamically share reward with neighbors so as to encourage agents to cooperate on the global objective through collectives. For each agent, the high-level policy learns how to share reward with neighbors to decompose the global objective, while the low-level policy learns to optimize the local objective induced by the high-level policies in the neighborhood. The two policies form a bi-level optimization and learn alternately. We empirically demonstrate that LToS outperforms existing methods in both social dilemma and networked MARL scenarios across scales.

## 1   Introduction

In fully cooperative multi-agent reinforcement learning (MARL), there are multiple agents interacting with the environment via their joint action to cooperatively optimize an objective. Many methods of centralized training and decentralized execution (CTDE) have been proposed for cooperative MARL, such as COMA (Foerster et al., 2018), QMIX (Rashid et al., 2018), QPLEX (Wang et al., 2021), and FOP (Zhang et al., 2021). However, these methods suffer from the overgeneralization issue: employed value functions cannot estimate well because agents sometimes choose uncoordinated actions, and thus the optimal policy cannot be learned (Castellini et al., 2019). Moreover, they may not easily scale up with the number of agents due to centralized learning (Qu et al., 2020a).

In many MARL applications, there are a large number of agents that are deployed as a partially connected network and collaboratively make decisions to optimize the globally averaged return, such as communication networks (Kim et al., 2019) and traffic signal control (Wei et al., 2019). To deal with such scenarios, networked MARL is formulated to decompose the dependency among all agents into dependencies between only neighbors. To avoid decision-making with insufficient information, agents are permitted to exchange messages with neighbors over the network. In such settings, it is feasible for agents to learn to make decisions in a decentralized way (Zhang et al., 2018; Qu et al., 2020b). However, there are still difficulties of dependency if anyone attempts to make decisions independently, *e.g., prisoner's dilemma* and *tragedy of the commons* (Pérolat et al., 2017). Existing methods tackle these problems by consensus update of value function (Zhang et al., 2018), credit assignment (Wang et al., 2020), or reward shaping (Chu et al., 2020). However, these methods rely on

---

[†]Corresponding Author

36th Conference on Neural Information Processing Systems (NeurIPS 2022).

either access to the global state and joint action (Zhang et al., 2018) or hand-crafted reward functions (Wang et al., 2020; Chu et al., 2020).

Inspired by the fact that *sharing* plays a key role in human's learning of cooperation (Eisenberg and Mussen, 1989), we propose ***learning to share*** (**LToS**), a hierarchically decentralized learning framework for networked MARL. LToS enables agents to learn to dynamically share reward with neighbors so as to collaboratively optimize the global objective. The high-level policies decompose the global objective into local ones by determining how to share their rewards, while the low-level policies optimize local objectives induced by the high-level policies. LToS learns in a decentralized manner, and we prove that the high-level policies are a mean-field approximation of the joint high-level policy. Moreover, the high-level and low-level policies form a bi-level optimization and alternately learn to optimize the global objective.

LToS is a general hierarchical framework for networked MARL and can be easily realized by diverse combinations of RL algorithms. We currently implement LToS by DDPG (Lillicrap et al., 2016) as the high-level policy and DGN (Jiang et al., 2020) as the low-level policy. We empirically demonstrate that LToS outperforms existing methods for networked MARL in both social dilemma and networked MARL scenarios.

## 2 Related Work

There are many recent studies for collaborative MARL. Most adopt CTDE (Rashid et al., 2018; Wang et al., 2021; Zhang et al., 2021; Su and Lu, 2022). Many of them are constructed on the basis of factorizing the joint Q-function (Rashid et al., 2018; Wang et al., 2021). However, these factorized methods suffer from the overgeneralization issue (Castellini et al., 2019). Other studies focus more on *decentralized training*, to which our work is more closely related, as summarized as follows.

**Networked MARL.** Zhang et al. (2018) and Qu et al. (2019) proposed consensus update of local value functions, where each agent keeps a local copy of the global value function but is assumed to have global information. Qu et al. (2020a) proposed intention propagation between agents, where each agent updates its policy based on intentions shared by other agents, but the policy may converge slowly due to propagated intentions over the network. Qu et al. (2020b) and Lin et al. (2020) investigated the exponential decay property, *i.e.*, the impact of agents on each other decays exponentially in their graph distance, while Chu et al. (2020) introduced a spatial discount factor to capture the influence between agents, which remains hand-tuned. However, none of these studies provide an explicit mechanism to solve social dilemmas in networked MARL.

**Reward Design.** Hostallero et al. (2020) aimed at maximizing social welfare, but they simply used temporal difference error for reward shaping. As temporal difference error in deep RL hardly converges to zero, it still biases the optimization objective. Mguni et al. (2019) added an extra part to the original reward as non-potential based reward shaping and used Bayesian optimization to induce the convergence to a desirable equilibrium between agents. However, the extra part remains fixed during an episode, which makes it less capable of dealing with dynamic environments. Moreover, the reward shaping alters the original optimization problem. Hughes et al. (2018) proposed the inequity aversion model to balance agents' selfish desire and social fairness. Wang et al. (2020) considered learning the Shapley value as the credit assignment. However, these methods still rely on hand-crafted reward designs. Lupu and Precup (2020) added *gifting* as an extra action into the original MDP to modify the MDP and objective. Yang et al. (2020) proposed that each agent learns an incentive function and optimizes the policy in terms of both reward and incentives given by other agents. Obviously, both methods alter the original objective of optimization.

Unlike existing work, LToS enables agents to learn to dynamically share reward with other agents *without the bias of the optimization objective* such that they can collaboratively optimize the global objective in networked MARL.

## 3 Background

### 3.1 Networked Multi-Agent Reinforcement Learning

Assume $N$ agents interact with an environment. Let $\mathcal{V} = \{1, 2, \cdots, N\}$ be the set of agents. The multi-agent system is modeled as an undirected graph $\mathcal{G}(\mathcal{V}, \mathcal{E})$, where each agent $i$ serves as vertex

$i$ and $\mathcal{E} \subseteq \mathcal{V} \times \mathcal{V}$ is the set of all edges. Two agents $i, j \in \mathcal{V}$ can communicate with each other if and only if $e_{ij} = (i, j) \in \mathcal{E}$. We denote *agent $i$ and its all neighbors in the graph together as a set* $\mathcal{N}_i$. The state of the environment $s \in \mathcal{S}$ transitions upon joint action $\boldsymbol{a} \in \mathcal{A}$ according to transition probability $\mathcal{P}_a : \mathcal{S} \times \mathcal{A} \times \mathcal{S} \to [0, 1]$, where joint action set $\mathcal{A} = \times_{i \in \mathcal{V}} \mathcal{A}_i$. Each agent $i$ has a policy $\pi_i \in \Pi_i : \mathcal{S} \times \mathcal{A}_i \to [0, 1]$, and we denote the joint policy of all agents as $\boldsymbol{\pi} \in \Pi = \times_{i \in \mathcal{V}} \Pi_i$ (Zhang et al., 2018). For networked MARL, a common and realistic assumption is that the reward of each agent $i$ just depends on its action and the actions of its neighbors (Qu et al., 2020a), *i.e.*, $r_i(s, \boldsymbol{a}) = r_i(s, a_{\mathcal{N}_i})$. Moreover, each agent $i$ may only obtain partial observation $o_i \in \mathcal{O}_i$, but can approximate the state by the observations of $\mathcal{N}_i$ (Jiang et al., 2020) or the observation history (Chu et al., 2020), which are all denoted by $o_i$ for *simplicity*. The *global objective* is to maximize the sum of cumulative rewards of all agents , *i.e.*, $\sum_{t=0}^{\infty} \sum_{i=1}^{N} \gamma^t r_i^t$.

## 3.2 Markov Game

In such a setting, each agent could individually maximize its own expected return, which is known as the Markov game. This may lead to stable outcome or Nash equilibrium, which however is usually sub-optimal in terms of the global objective. Given the joint policy $\boldsymbol{\pi}$, the value function of agent $i$ is given by

$$v_i^{\boldsymbol{\pi}}(s) = \sum_{\boldsymbol{a}} \boldsymbol{\pi}(\boldsymbol{a}|s) \sum_{s'} p_a(s'|s, \boldsymbol{a})[r_i + \gamma v_i^{\boldsymbol{\pi}}(s')], \tag{1}$$

where $p_a \in \mathcal{P}_a$ describes the state transitions. Then, a Nash equilibrium is defined as (Mguni et al., 2019)

$$v_i^{(\pi_i, \pi_{-i})}(s) \geq v_i^{(\pi_i', \pi_{-i})}(s), \quad \forall \pi_i' \in \Pi_i, \forall s \in \mathcal{S}, \forall i \in \mathcal{V},$$

where $\pi_{-i} = \times_{j \in \mathcal{V} \setminus \{i\}} \pi_j$.

# 4 Method

LToS is a decentralized hierarchy. At each agent, the high-level policy determines the weights of reward sharing based on low-level policies while the low-level policy directly interacts with the environment to optimize the local objective induced by the high-level policies. Therefore, they form a bi-level optimization and alternately learn towards the global objective.

## 4.1 Reward Sharing

The intuition of reward sharing is that if agents share their rewards with others, each agent has to consider the consequence of its actions on others, and thus it promotes cooperation. In networked MARL, as the reward of an agent is assumed to depend on the actions of neighbors, we allow reward sharing only between neighboring agents. This is because the change of actions of neighbors directly affects the reward while the agents outside the neighborhood can only affect the return of the agent indirectly by the change of state distribution. Moreover, this also fits the setting of networked MARL with restricted communication between neighbors.

For the graph of $\mathcal{V}$, we additionally define a set of directed edges, $\mathcal{D}$, constructed from $\mathcal{E}$. Specifically, we add a loop $d_{ii} \in \mathcal{D}$ for each agent $i$ and split each undirected edge $e_{ij} \in \mathcal{E}$ into two directed edges: $d_{ij} = (i, j)$ and $d_{ji} = (j, i) \in \mathcal{D}$. Each agent $i$ determines a weight $w_{ij} \in [0, 1]$ for each directed edge $d_{ij}, \forall j \in \mathcal{N}_i$, subject to the constraint $\sum_{j \in \mathcal{N}_i} w_{ij} = 1$, so that $w_{ij}$ proportion of agent $i$'s environment reward $r_i$ will be shared to agent $j$. Let $\boldsymbol{w} \in \mathcal{W} = \times_{d_{ij} \in \mathcal{D}} w_{ij}$ be the weights of the graph. Therefore, the shaped reward after sharing for each agent $i$ is defined as

$$r_i^{\boldsymbol{w}} = \sum_{j \in \mathcal{N}_i} w_{ji} r_j. \tag{2}$$

## 4.2 Hierarchy

Assume there is a joint high-level policy $\phi \in \Phi : \mathcal{S} \times \mathcal{W} \to [0, 1]$ to determine $\boldsymbol{w}$. Given $\phi$ and $\boldsymbol{w}$, we can define the value function of $\boldsymbol{\pi}$ at each agent $i$ based on (1) as

$$v_i^{\boldsymbol{\pi}}(s; \phi) = \sum_{\boldsymbol{w}} \phi(\boldsymbol{w}|s) \sum_{\boldsymbol{a}} \boldsymbol{\pi}(\boldsymbol{a}|s, \boldsymbol{w}) \sum_{s'} p_a(s'|s, \boldsymbol{a})[r_i^{\boldsymbol{w}} + \gamma v_i^{\boldsymbol{\pi}}(s'; \phi)], \tag{3}$$

$$v_i^{\boldsymbol{\pi}}(s; \boldsymbol{w}, \phi) = \sum_{\boldsymbol{a}} \boldsymbol{\pi}(\boldsymbol{a}|s, \boldsymbol{w}) \sum_{s'} p_a(s'|s, \boldsymbol{a})[r_i^{\boldsymbol{w}} + \gamma v_i^{\boldsymbol{\pi}}(s'; \phi)].$$

It is noteworthy that $\boldsymbol{w}$ is a multidimensional action for an allocation scheme rather than a probability distribution. In our derivation, we express $\boldsymbol{w}$ as a discrete action for simplicity. It also holds for continuous action as long as we change all the summations to integrals. Let $V_{\mathcal{V}}^{\phi}(s; \boldsymbol{\pi}) \doteq \sum_{i \in \mathcal{V}} v_i^{\boldsymbol{\pi}}(s; \phi)$ and $Q_{\mathcal{V}}^{\phi}(s, \boldsymbol{w}; \boldsymbol{\pi}) \doteq \sum_{i \in \mathcal{V}} v_i^{\boldsymbol{\pi}}(s; \boldsymbol{w}, \phi)$.

**Proposition 4.1.** *Given $\boldsymbol{\pi}$, $V_{\mathcal{V}}^{\phi}(s; \boldsymbol{\pi})$ and $Q_{\mathcal{V}}^{\phi}(s, \boldsymbol{w}; \boldsymbol{\pi})$ are respectively the value function and action-value function of $\phi$.*

*Proof.* The proof is deferred to Appendix A. □

Proposition 4.1 implies that $\phi$ directly optimizes the global objective by generating $\boldsymbol{w}$, given $\boldsymbol{\pi}$. Unlike existing hierarchical RL methods, we can directly construct the value function and action-value function of $\phi$ based on the value function of $\boldsymbol{\pi}$ at each agent.

As $\phi$ optimizes the global objective given $\boldsymbol{\pi}$ while $\pi_i$ optimizes the shaped reward individually at each agent given $\phi$ (assuming $\boldsymbol{\pi}$ convergent to Nash equilibrium or stable outcome, denoted as lim), they form a bi-level optimization. Let $J_{\phi}(\boldsymbol{\pi})$ and $J_{\boldsymbol{\pi}}(\phi)$ denote the objectives of $\phi$ and $\boldsymbol{\pi}$ respectively. The bi-level optimization can be formulated as follows,

$$\max_{\phi} \quad J_{\phi}(\boldsymbol{\pi}^*(\phi))$$
$$s.t. \quad \boldsymbol{\pi}^*(\phi) = \arg \lim_{\boldsymbol{\pi}} J_{\boldsymbol{\pi}}(\phi). \tag{4}$$

## 4.3 Decentralized Learning

We start from collective learning to achieve global optimization of average reward. So far, the joint high-level policy is still in a centralized form. Note that the scenario needs a decentralized method and each agent has its own reward. Now we turn to learning the joint high-level policy in a decentralized way. Let $w_i^{\text{out}} \doteq \{w_{ij}|j \in \mathcal{N}_i\}$ and $w_i^{\text{in}} \doteq \{w_{ji}|j \in \mathcal{N}_i\}$. The following proposition proves each agent's independence of each other on the high level.

**Proposition 4.2.** *The joint high level policy $\phi$ can be learned in a decentralized manner, and the decentralized high-level policies of all agents form a mean-field approximation of $\phi$.*

*Proof.* The proof is deferred to Appendix A. □

Proposition 4.1 and 4.2 indicate that for each agent $i$, the low-level policy simply learns a local $\pi_i(a_i|s, w_i^{\boldsymbol{w}})$ to optimize the cumulative reward of $r_i^{\boldsymbol{w}}$, since $r_i^{\boldsymbol{w}}$ is fully determined by $w_i^{\text{in}}$ according to (2) and denoted as $r_i^w$ from now on. And the high-level policy $\phi_i$ just needs to locally determine $w_i^{\text{out}}$ to optimize the cumulative reward of $r_{\mathcal{V}}^{\phi}$.

Therefore, for decentralized learning, (4) can be decomposed locally for each agent $i$ as

$$\max_{\phi_i} \quad J_{\phi_i}(\phi_{-i}, \pi_1^*(\phi), \cdots, \pi_N^*(\phi))$$
$$s.t. \quad \pi_i^*(\phi) = \arg \max_{\pi_i} J_{\pi_i}(\pi_{-i}, \phi_1(\boldsymbol{\pi}), \cdots, \phi_N(\boldsymbol{\pi})). \tag{5}$$

Now we use max instead of lim because local policies can be compared and improved in a decentralized manner in a Markov game. We abuse the notation and let $\phi$ and $\pi$ also denote their parameterizations respectively. To solve the optimization, we have

$$\nabla_{\phi_i} J_{\phi_i}(\phi_{-i}, \pi_1^*(\phi), \cdots, \pi_N^*(\phi))$$
$$\approx \nabla_{\phi_i} J_{\phi_i}(\phi_{-i}, \pi_1 + \alpha \nabla_{\pi_1} J_{\pi_1}(\phi), \cdots, \pi_N + \alpha \nabla_{\pi_N} J_{\pi_N}(\phi)), \tag{6}$$

where $\alpha$ is the learning rate for the low-level policy. Let $\pi_i'$ denote $\pi_i + \alpha \nabla_{\pi_i} J_{\pi_i}(\boldsymbol{\phi})$, we have

$$\nabla_{\phi_i} J_{\phi_i}(\phi_{-i}, \pi_1^*(\boldsymbol{\phi}), \cdots, \pi_N^*(\boldsymbol{\phi}))$$

$$\approx \nabla_{\phi_i} J_{\phi_i}(\phi_{-i}, \pi_1', \cdots, \pi_N') + \alpha \sum_{j=1}^{N} \nabla_{\phi_i, \pi_j}^2 J_{\pi_j}(\boldsymbol{\phi}) \nabla_{\pi_j'} J_{\phi_i}(\phi_{-i}, \pi_1', \cdots, \pi_N').$$

The second-order derivative is neglected due to high computational complexity, without incurring significant performance drop such as in meta-learning (Finn et al., 2017) and neural architecture search (Liu et al., 2019). Differently, our low-level policy requires more than one gradient step until convergence. Similarly, we have

$$\nabla_{\pi_i} J_{\pi_i}(\pi_{-i}, \phi_1^*(\boldsymbol{\pi}), \cdots, \phi_N^*(\boldsymbol{\pi}))$$

$$\approx \nabla_{\pi_i} J_{\pi_i}(\pi_{-i}, \phi_1 + \beta \nabla_{\phi_1} J_{\phi_1}(\boldsymbol{\pi}), \cdots, \phi_N + \beta \nabla_{\phi_N} J_{\phi_N}(\boldsymbol{\pi})),$$

where $\beta$ is the learning rate of the high-level policy. Therefore, we can solve the bi-level optimization (4) by the first-order approximations in a decentralized way. For each agent $i$, $\phi_i$ and $\pi_i$ are alternately updated.

In distributed learning, as each agent $i$ usually does not have access to state, we further approximate $\phi_i(w_i^{\text{out}}|s)$ and $\pi_i(a_i|s, w_i^{\text{in}})$ by $\phi_i(w_i^{\text{out}}|o_i)$ and $\pi_i(a_i|o_i, w_i^{\text{in}})$, respectively. Moreover, in network MARL as each agent $i$ is closely related to neighboring agents, (5) can be further seen as $\pi_i$ maximizes the cumulative discounted reward of $r_i^w$ given $\phi_{\mathcal{N}_i}$, where $\phi_{\mathcal{N}_i} = \times_{j \in \mathcal{N}_i} \phi_j$, and $\phi_i$ equivalently optimizes the global objective given $\pi_{\mathcal{N}_i}$, where $\pi_{\mathcal{N}_i} = \times_{j \in \mathcal{N}_i} \pi_j$. During training, $\pi_{\mathcal{N}_i}$ and $\phi_{\mathcal{N}_i}$ are implicitly considered by interactions of $w_i^{\text{out}}$ and $w_i^{\text{in}}$ respectively. The architecture of LToS is illustrated in Figure 1. At each timestep, the high-level policy of each agent $i$ makes a decision of action $w_i^{\text{out}}$ as the weights of reward sharing based on the observation. Then, the low-level policy takes the observation and $w_i^{\text{in}}$ as an input and outputs the action. Agent $i$ ob-

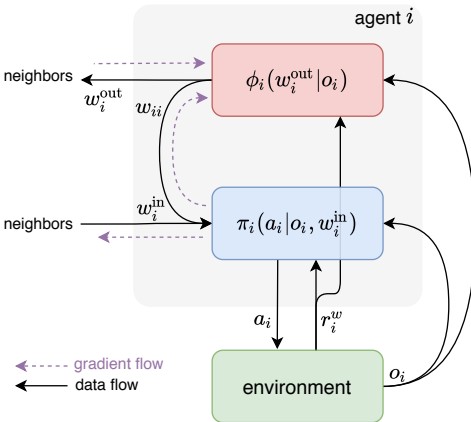

Figure 1: LToS

tains the shaped reward according to $w_i^{\text{in}}$ for both the high-level and low-level policies. The gradients are backpropagated along purple dotted lines.

Further, from Proposition 4.1, we have: $q_i^{\phi_i}(s, w_i^{\text{out}}; \pi_{\mathcal{N}_i}) = v_i^{\pi_i}(s; w_i^{\text{in}}, \phi_{\mathcal{N}_i})$, where $q_i^{\phi_i}$ is the action-value function of $\phi_i$ given $\pi_{\mathcal{N}_i}$, $v_i^{\pi_i}$ is the value function of $\pi_i$ given $\phi_{\mathcal{N}_i}$ and conditioned on $w_i^{\text{in}}$. As aforementioned, we approximately have $q_i^{\phi_i}(o_i, w_i^{\text{out}}) = v_i^{\pi_i}(o_i; w_i^{\text{in}})$. We can see that the action-value function of $\phi_i$ is equivalent to the value function of $\pi_i$. That said, we can use a single network to approximate these two functions simultaneously. For a deterministic low-level policy, the high-level and low-level policies can share the same action-value function. In the current instantiation of LToS, we use DGN (Jiang et al., 2020) (Q-learning) for the low-level policy and DDPG (Lillicrap et al., 2016) for the high-level policy. Thus, the Q-network of DGN also serves as the critic of DDPG, and the gradient of $w_i^{\text{in}}$ is calculated based on the maximum Q-value of $a_i$.

For completeness, Algorithm 1 (see Appendix B) gives the training procedure of LToS based on DDPG and DGN. More discussions about training LToS are also available in Appendix C. The code of LToS is available at https://github.com/PKU-RL/RoadnetSZ.

## 5   Experiments

For the experiments, we adopt three scenarios *prisoner*, *jungle*, and *traffic* depicted in Figure 2, where *prisoner* and *jungle* (Jiang et al., 2020) are grid games about social dilemma that easily measures agents' cooperation, while *traffic* is a realistic scenario of networked MARL. We obey the principle of networked MARL that only allows communication in neighborhood as Zhang et al. (2018) and Chu et al. (2020).

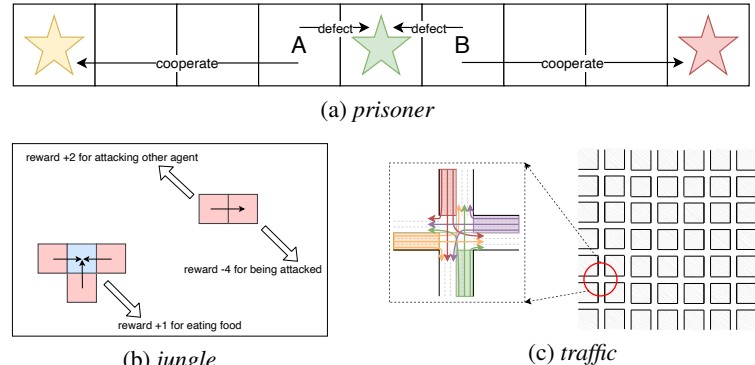

(a) *prisoner*

(b) *jungle*
(c) *traffic*

Figure 2: Three experimental scenarios: (a) *prisoner*, (b) *jungle*, and (c) *traffic*.

To illustrate the reward sharing scheme each agent learned, we use a simple indicator: *selfishness*, the reward proportion that an agent chooses to keep for itself. For ablation, we keep the sharing weights fixed for each agent, named *fixed* LToS. Throughout the experiments, we additionally compare with the baselines including DQN and DGN, where DGN also serves the ablation of LToS without reward sharing as DGN is the low-level policy of LToS. To maximize the global return directly by centralized learning, we use QMIX (Rashid et al., 2018) as a baseline throughout the three scenarios and two other ones in *prisoner*. Moreover, as LToS aims to bring harmonious cooperation by reward sharing in networked MARL, we compared LToS to three methods for networked MARL, *i.e.*, ConseNet (Zhang et al., 2018), NeurComm (Chu et al., 2020) and Intention Propagation (abbreviated as IP) (Qu et al., 2020a), and LIO (Yang et al., 2020) for incentivized learning, all of which use recurrent neural network (RNN) or graph neural network (GNN) for the partially observable environment. More details of hyperparameters are available in Appendix D.

## 5.1 Prisoner

We use *prisoner*, a grid game version of the well-known matrix game *prisoner's dilemma* from Sodomka et al. (2013) to empirically demonstrate that LToS is able to learn cooperative policies to achieve the global optimum (*i.e.*, maximize globally averaged return). As illustrated in Figure 2a, there are two agents $A$ and $B$ that respectively start on two sides of the middle of a grid corridor with *full* observation. At each timestep, each agent chooses an action *left* or *right* and moves to the corresponding adjacent grid, and every action incurs a cost $-0.01$. There are three goals, two goals at both ends and one in the middle. The agent gets a reward $+1$ for reaching the goal. The game ends once some agent reaches a goal or two agents reach different goals simultaneously. This game resembles prisoner's dilemma: going for the middle goal ("defect") will bring more rewards than the farther one on its side ("cooperate"), but if two agents both adopt that, a collision occurs and only one of the agents wins the goal with equal probability. On the contrary, both agents obtain a higher return if they both "cooperate", though it takes more steps. The highest possible return is $1$.

Figure 3a illustrates the learning curves of all the methods in terms of average return. Note that for all three scenarios, we present the average of 5 training runs with different random seeds by solid lines and the min/max value by shadowed areas. As a result of self-interest optimization, DQN converges to the "defect/defect" Nash equilibrium where each agent receives an expected reward about $0.5$. So does DGN since it only aims to take advantage of its neighbors' observations while *prisoner* is a fully observable environment already. Given a hand-tuned reward shaping factor to direct agents to maximize average return, NeurComm and *fixed* LToS agents are able to cooperate eventually. So are ConseNet and QMIX. However, they converge slowly. In contrast, IP agents learn at a slower pace and its performance is only a little higher than $0.5$. LIO agents cooperate soon enough at the beginning, but they cannot form steady cooperation and perhaps need longer training to get rid of such instability.

JointDQN, Coco-Q (Sodomka et al., 2013), and LToS perform similarly and outperform other methods. JointDQN is one centralized DQN that takes control of joint actions of both agent $A$ and $B$, and thus should be able to achieve the best performance but still takes time to converge even in such a simple two-agent scenario. As a modified tabular Q-learning method, Coco-Q introduces the coco

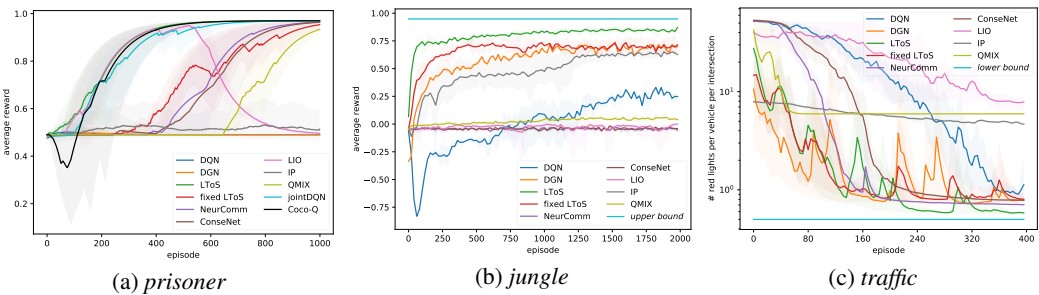

Figure 3: Learning curves in (a) *prisoner*, (b) *jungle*, and (c) *traffic*. All the curves are plotted using 5 training runs with different random seeds, where the solid line is the mean and the shadowed area is enclosed by the min and max value.

value (Kalai and Kalai, 2010) as a substitute for the expected return in the Bellman equation and regards the difference as transferred reward. However, it is specifically designed for some games, and it is hard to be extended beyond two-player games. LToS can learn the reward sharing scheme where one agent at first gives all the reward to the other so that both of them are prevented from "defect", and thus achieve the best average return quickly, as observed in the experiment. By *prisoner*, we verify that LToS can escape from local optimum by learning to share reward.

## 5.2   Jungle

*Jungle* is a scenario about moral dilemma proposed by Jiang et al. (2020) based on MAgent (Zheng et al., 2017). As illustrated in Figure 2b, there are $N$ agents and $L$ stationary foods. At each timestep, each agent can attack or move to one adjacent grid. Eating (attacking food) brings a positive reward $+1$, but attacking other agents obtains a higher reward $+2$. The victim, however, suffers a negative reward $-4$, which makes each attack between agents a negative-sum action. Moreover, attacking a blank grid gets a small negative reward $-0.01$ (inhibiting excessive attacks). We follow the original setting of Jiang et al. (2020): map size = $30 \times 30$ grids, $N = 20$, $L = 12$, and the observation consists of one's coordinates and a field of $11 \times 11$ grids nearby. Each agent has 3 closest agents as its neighbors. Compared to *prisoner*, *jungle* has much more agents, and thus JointDQN and Coco-Q are disregarded for this scenario. Another challenge of *jungle* is that the network topology is dynamic since each agent can always move. Fortunately, the topology change slowly and predictably, and algorithms may get the time-varying neighbor set $\mathcal{N}_i$ as part of the input. Therefore, it is still likely to estimate the shaped rewards and value functions well.

Table 1: Average reward per step of all the methods in *jungle*.

| DQN | DGN | *fixed* LToS | **LToS** | NeurComm | ConseNet | LIO | IP | QMIX | *upper bound* |
|------|------|------|------|------|------|------|------|------|------|
| 0.24 | 0.66 | 0.71 | **0.86** | -0.05 | -0.04 | 0.00 | 0.63 | 0.04 | 0.95 |

Figure 3b illustrates the learning curves of all the methods, and their performance after convergence is also summarized in Table 1. NeurComm, ConseNet and QMIX do not perform well in this task. In NeurComm, each agent gets a "delayed global information". However, a stable pattern of delayed global information cannot be formed when the communication is conducted via a dynamic topology. ConseNet is constructed on the basis of a premise of *full observation*, and it can hardly learn well when the input is not only partial but also fairly varying in sequence and content. LIO agents also perform badly, since they cannot be distinguished from one another in the dynamic topology and thus fail to learn a proper incentive function. Moreover, LIO requires opponent modeling, but it is hard to simultaneously model all other agents in a dynamic environment. QMIX is free from these problems. While aiming at global optimization, like LIO, it realizes that attacking usually means a negative-sum action, but as a result, it avoids attacking as well as eating most of the time and thus only achieves a reward slightly higher than 0. Another possible reason to explain the performance QMIX is its scalability. As there are 20 agents in the scenario, it can be hard to learn the joint action-value function to directly optimize the average return (Qu et al., 2020a). Also, we can see that a fixed reward sharing scheme does not bring any gain over DGN. This is because fixed reward sharing does

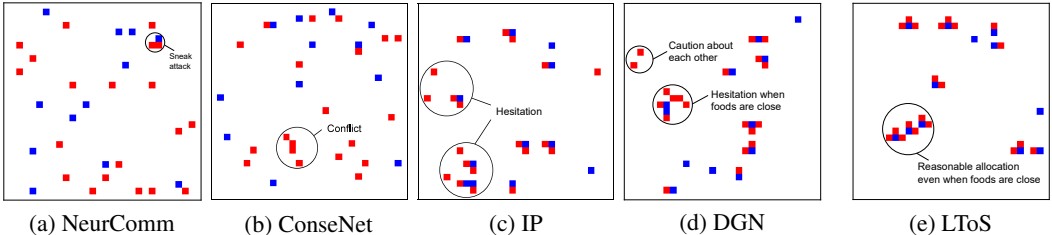

| (a) NeurComm | (b) ConseNet | (c) IP | (d) DGN | (e) LToS |

Figure 4: Representative behaviors of agents learned by (a) NeurComm, (b) ConseNet, (c) IP, (d) DGN, and (e) LToS in *jungle*.

not adapt to the dynamic topology. By learning proper reward sharing and adjusting to changing circumstances, LToS outperforms all other baselines. Note that there is an upper bound for average reward per step for *jungle*. By estimating the average distance between each agent and the food that is the closest to it at the beginning of one episode, we can give a loose upper bound around $0.95$ to reflect our improvement.

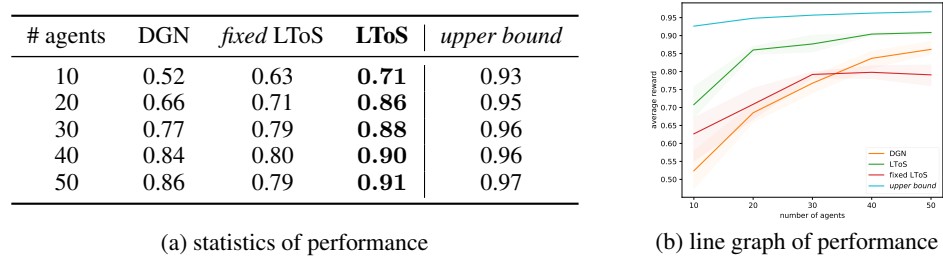

| # agents | DGN | *fixed* LToS | **LToS** | *upper bound* |
|----------|------|------------|----------|-------------|
| 10 | 0.52 | 0.63 | **0.71** | 0.93 |
| 20 | 0.66 | 0.71 | **0.86** | 0.95 |
| 30 | 0.77 | 0.79 | **0.88** | 0.96 |
| 40 | 0.84 | 0.80 | **0.90** | 0.96 |
| 50 | 0.86 | 0.79 | **0.91** | 0.97 |

(a) statistics of performance

(b) line graph of performance

Figure 5: Average reward per step of methods in *jungle* with different number of agents.

Figure 4 illustrates the representative behaviors of agents learned by difference methods. For NeurComm, ConseNet (and the same with LIO), most agents are not even close to the foods, so as to avoid being attacked. Even though, there are still conflict and sneak attack between agents sometimes. IP and DGN agents learn much better, but agents may still be cautious about each other, which leads to hesitation when they are near the same food. LToS agents learn to properly share the food even if the foods are close (*i.e.*, agents are easy to be attacked) as depicted in Figure 4e and demonstrate much better cooperation than the agents learned by other methods. The experimental results in *jungle* verify that LToS can also adapts to considerably varying topology in networked MARL.

Besides, we compared LToS and ablation baselines (*i.e.*, DGN and *fixed* LToS) with different number of agents (*i.e.*, from 10 to 50) to verify the scalability of LToS. All the setting remains the same except that the number of agents and food grows proportionally (*i.e.*, $\#agents/\#foods = 5/3$). As depicted in Figure 5, LToS can always achieve the best performance as the agent population size increases.

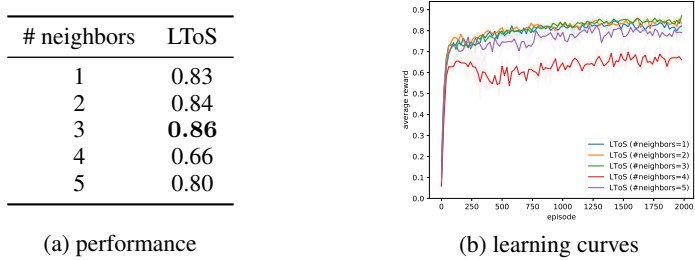

| # neighbors | LToS |
|-------------|------|
| 1 | 0.83 |
| 2 | 0.84 |
| 3 | **0.86** |
| 4 | 0.66 |
| 5 | 0.80 |

(a) performance

(b) learning curves

Figure 6: Average reward per step of LToS in *jungle* with different number of neighbors.

We also conducted a study on the impact of graph density. To be specific, we run LToS with different number of neighbors (*i.e.*, from 1 to 5). As illustrated is Figure 6, the number of neighbors indeed affects the performance. With the increase of neighbors, the performance increases first and then decreases in an indefinitive way. We think the reason may be that when the graph density increases,

the learning difficulty also increases as the agent needs to decide how to share rewards with more neighbors. Beyond a certain threshold, the learning difficulty outweighs the gain. This is similar to learning to communicate, where the graph density increases, the gain of communication decreases due to learning difficulty (Ding et al., 2020). However, as in networked MARL the graph structure is typically assumed to be sparse and given, this should not be a problem.

## 5.3   Traffic

In *traffic*, as illustrated in Figure 2c , we aim to investigate the capability of LToS in dealing with highly dynamic environment through reward sharing. We adopt the same problem setting as Wei et al. (2019). In a road network, each agent serves as traffic signal control at an intersection. The observation of an agent consists of a one-hot representation of its current phase (directions for red/green lights) and the number of vehicles on each incoming lane of the intersection. At each timestep, an agent chooses a phase from the pre-defined phase set for the next time interval, *i.e.*, 10 seconds. The reward is set to be the

Table 2: Statistics of traffic flows

| Time (second) | Arrival Rate (vehicles/s) |
|---|---|
| $0 - 600$ | 1 |
| $600 - 1,200$ | 1/4 |
| $1,200 - 1,800$ | 1/3 |
| $1,800 - 2,400$ | 2 |
| $2,400 - 3,000$ | 1/5 |
| $3,000 - 3,600$ | 1/2 |

negative of the sum of the queue lengths of all approaching lanes at current timestep. The global objective is to minimize average wait time of all vehicles in the road network, which is equivalent to minimizing the sum of queue lengths of all intersections over an episode (Zheng et al., 2019). The experiment was conducted on a traffic simulator, CityFlow (Zhang et al., 2019). We use a $6 \times 6$ grid network with 36 intersections. The traffic flows were generated to simulate dynamic traffic flows including both peak and off-peak period, and the statistics are summarized in Table 2.

Table 3: Average number of red lights one vehicle waits for at per intersection of methods in *traffic*.

| Network | DQN | DGN | *fixed* LToS | **LToS** | NeurComm | ConseNet | LIO | IP | QMIX | *lower bound* |
|---|---|---|---|---|---|---|---|---|---|---|
| $6 \times 6$ | 0.90 | 0.78 | 0.80 | **0.58** | 0.71 | 0.78 | 7.74 | 4.67 | 5.94 | 0.50 |
| Shenzhen | 13.99 | 2.02 | 1.94 | **1.71** | 2.27 | 7.63 | 19.15 | 5.52 | 19.74 | 1.50 |

For better demonstration, we choose to show the normalized metric of wait time: the average number of red lights one vehicle waits for at per intersection, and we can also give a loose lower bound 0.50 to reflect our improvement. Figure 3c shows the learning curves of all the methods in terms of that in logarithmic form. The performance after convergence is summarized in Table 3, where LToS outperforms all other methods. LToS outperforms DGN, which demonstrates the reward sharing scheme learned by the high-level policy indeed helps to improve the cooperation of agents. Without the high-level policy, *i.e.*, given fixed sharing weights, *fixed* LToS does not perform well in dynamic environment. This indicates the necessity of the high-level policy. The performance of IP agents increases so slowly that they cannot converge efficiently. Although NeurComm and ConseNet both take advantage of RNN for partially observable environments, LToS still outperforms these methods, which verifies the great improvement of LToS in networked MARL. QMIX is confined to suboptimality (Mahajan et al., 2019). As observed in the experiment, QMIX tries to release traffic flows from one direction while stopping flows from the other direction all the time, because this will only make two rows of intersections on the border blocked but keep most of the intersections from any traffic jam all the time. However, the global optimality actually does not need to be constructed on the sacrifice of anyone. Some similar thing happens to LIO. It is likely because LIO contains some sensitive parameters (Yang et al., 2020) and agents are hard to learn and coordinate their incentive functions since the original reward functions change acutely once an improper operation causes traffic congestion. An introduction of some explicit coordination mechanism may also alleviate the problem, like that of NeurComm and ConseNet.

We visualize the variation of *selfishness* of all agents during an episode in *traffic* in Figure 7a and 7b. Figure 7a depicts the temporal variance of selfishness for each agent. For most agents, there are two valleys occurred exactly during two peak periods (*i.e.*, $0 - 600$s and $1,800 - 2,400$s). This is because for heavy traffic agents need to cooperate more closely, which can be induced by being less selfish. We can see this from the fact that selfishness is even lower in the second valley where the traffic is even heavier (*i.e.*, 2 *vs.* 1 vehicles/s). Therefore, this demonstrates that the agents learn to adjust their extent of cooperation to deal with dynamic environment by controlling the sharing weights.

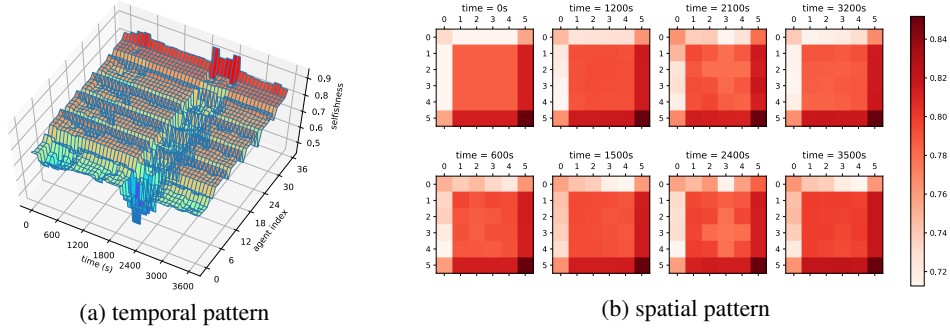

(a) temporal pattern               (b) spatial pattern

Figure 7: Patterns of *selfishness* in *traffic*.

Figure 7b shows the spatial pattern of selfishness at different timesteps, where the distribution of agents is the same as the road network in Figure 2c. The edge and inner agents tend to have very different selfishness. In addition, inner agents keep their selfishness more uniform during off-peak periods, while they diverge and present cross-like patterns during peak periods. This shows that handling heavier traffic requires more diverse reward sharing schemes among agents to promote more sophisticated cooperation.

In addition, we use another road network that is part of Shenzhen, China with 33 intersections as illustrated in Figure 8, and one-hour real traffic flows (Xu et al., 2022). The performance is also summarized in Table 3. It is shown that LToS still outperforms the baselines by a large margin. The experimental results in *traffic* verify that LToS can also handle highly dynamic environment in networked MARL.

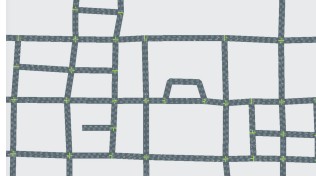

Figure 8: Shenzhen network

# 6 Conclusion

In this paper, we proposed LToS, a hierarchically decentralized framework for networked MARL. LToS enables agents to share reward with neighbors so as to encourage agents to cooperate on the global objective through collectives. For each agent, the high-level policy learns how to share reward with neighbors to decompose the global objective, while the low-level policy learns to optimize the local objective induced by the high-level policies in the neighborhood. Experimentally, we demonstrate that LToS outperforms existing methods in both social dilemma and networked MARL scenario across scales.

## Acknowledgments and Disclosure of Funding

This work is supported in part by NSF China under grant 61872009, Shenzhen Fundamental Research Program (GXWD 20201231165807007-202008806163656003), and Peng Cheng Lab. The authors would like to thank the anonymous reviewers for their valuable comments.

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
