# A Proofs

## A.1 Proof of Proposition 4.1

**Proposition 4.1.** *Given $\boldsymbol{\pi}$, $V_{\mathcal{V}}^{\phi}(s; \boldsymbol{\pi})$ and $Q_{\mathcal{V}}^{\phi}(s, \boldsymbol{w}; \boldsymbol{\pi})$ are respectively the value function and action-value function of $\phi$.*

*Proof.* Let $r_i^{\phi} \doteq \sum_{\boldsymbol{a}} \boldsymbol{\pi}(\boldsymbol{a}|s, \boldsymbol{w}) r_i^{\boldsymbol{w}}$ and $p_w(s'|s, \boldsymbol{w}) \doteq \sum_{\boldsymbol{a}} \boldsymbol{\pi}(\boldsymbol{a}|s, \boldsymbol{w}) p_a(s'|s, \boldsymbol{a})$ in contrast with $p_a$ . As commonly assumed the reward is deterministic given $s$ and $\boldsymbol{a}$, from (3), we have,

$$
\begin{aligned}
v_i^{\boldsymbol{\pi}}(s; \phi) &= \sum_{\boldsymbol{w}} \phi(\boldsymbol{w}|s) \sum_{\boldsymbol{a}} \boldsymbol{\pi}(\boldsymbol{a}|s, \boldsymbol{w})[r_i^{\boldsymbol{w}} + \sum_{s'} p_a(s'|s, \boldsymbol{a}) \gamma v_i^{\boldsymbol{\pi}}(s'; \phi)] \\
&= \sum_{\boldsymbol{w}} \phi(\boldsymbol{w}|s) \sum_{s'} p_w(s'|s, \boldsymbol{w})[r_i^{\phi} + \gamma v_i^{\boldsymbol{\pi}}(s'; \phi)],
\end{aligned} \tag{7}
$$

where $p_w \in \mathcal{P}_w : \mathcal{S} \times \mathcal{W} \times \mathcal{S} \to [0, 1]$ describes the state transitions given $\boldsymbol{\pi}$.

Let $r_{\mathcal{V}}^{\phi} \doteq \sum_{i \in \mathcal{V}} r_i^{\phi}$, and from (7) we have

$$
\begin{aligned}
V_{\mathcal{V}}^{\phi}(s; \boldsymbol{\pi}) &= \sum_{i \in \mathcal{V}} \sum_{\boldsymbol{w}} \phi(\boldsymbol{w}|s) \sum_{s'} p_w(s'|s, \boldsymbol{w})[r_i^{\phi} + \gamma v_i^{\boldsymbol{\pi}}(s'; \phi)] \\
&= \sum_{\boldsymbol{w}} \phi(\boldsymbol{w}|s) \sum_{s'} p_w(s'|s, \boldsymbol{w})[\sum_{i \in \mathcal{V}} r_i^{\phi} + \gamma \sum_{i \in \mathcal{V}} v_i^{\boldsymbol{\pi}}(s'; \phi)] \\
&= \sum_{\boldsymbol{w}} \phi(\boldsymbol{w}|s) \sum_{s'} p_w(s'|s, \boldsymbol{w})[r_{\mathcal{V}}^{\phi} + \gamma V_{\mathcal{V}}^{\phi}(s'; \boldsymbol{\pi})],
\end{aligned}
$$

and similarly,

$$
\begin{aligned}
Q_{\mathcal{V}}^{\phi}(s, \boldsymbol{w}; \boldsymbol{\pi}) &= \sum_{i \in \mathcal{V}} \sum_{s'} p_w(s'|s, \boldsymbol{w})[r_i^{\phi} + \gamma \sum_{\boldsymbol{w}'} \phi(\boldsymbol{w}'|s') v_i^{\boldsymbol{\pi}}(s'; \boldsymbol{w}', \phi)] \\
&= \sum_{s'} p_w(s'|s, \boldsymbol{w})[r_{\mathcal{V}}^{\phi} + \gamma \sum_{\boldsymbol{w}'} \phi(\boldsymbol{w}'|s') Q_{\mathcal{V}}^{\phi}(s', \boldsymbol{w}'; \boldsymbol{\pi})].
\end{aligned}
$$

Moreover, from the definitions of $r_i^{\boldsymbol{w}}$ and $r_i^{\phi}$ we have

$$
\begin{aligned}
r_{\mathcal{V}}^{\phi} &= \sum_{\boldsymbol{a}} \boldsymbol{\pi}(\boldsymbol{a}|s, \boldsymbol{w}) \sum_{i \in \mathcal{V}} r_i^{\boldsymbol{w}} \\
&= \sum_{\boldsymbol{a}} \boldsymbol{\pi}(\boldsymbol{a}|s, \boldsymbol{w}) \sum_{i \in \mathcal{V}} \sum_{j \in \mathcal{N}_i} w_{ji} r_j \\
&= \sum_{\boldsymbol{a}} \boldsymbol{\pi}(\boldsymbol{a}|s, \boldsymbol{w}) \sum_{(i,j) \in \mathcal{D}} w_{ij} r_i = \sum_{\boldsymbol{a}} \boldsymbol{\pi}(\boldsymbol{a}|s, \boldsymbol{w}) \sum_{i \in \mathcal{V}} r_i.
\end{aligned}
$$

Thus, given $\boldsymbol{\pi}$, $V_{\mathcal{V}}^{\phi}(s)$ and $Q_{\mathcal{V}}^{\phi}(s, \boldsymbol{w})$ are respectively the value function and action-value function of $\phi$ in terms of the sum of expected cumulative rewards of all agents, *i.e.*, the global objective. □

## A.2 Proof of Proposition 4.2

**Proposition 4.2.** *The joint high level policy $\phi$ can be learned in a decentralized manner, and the decentralized high-level policies of all agents form a mean-field approximation of $\phi$.*

First, we introduce one definition and one lemma.

**Definition 1** (**Markov Random Field**). *A Markov Random Field (MRF) is a graph $\mathcal{G} = (\mathcal{V}, \mathcal{E})$ that satisfies:*

$$
P(X_i | \{X_j\}_{j \in \mathcal{V} \setminus \{i\}}) = P(X_i | \{X_j\}_{j \in \mathcal{N}_i}) \tag{8}
$$

where $X_i$ is some random variable associated with node $i, \forall i \in \mathcal{V}$.

**Lemma A.1** (**Hammersley–Clifford Theorem**). *A probability distribution that has a strictly positive mass or density satisfies one of the Markov properties with respect to an undirected graph $\mathcal{G}$ if and only if it is a Gibbs random field, i.e., its density can be factorized over the cliques (or complete subgraphs) of the graph. (Hammersley and Clifford, 1971)*

Now we begin the proof of Proposition 4.2.

*Proof.* Let $d_{ij} \in \mathcal{D}$ serve as a vertex with action $w_{ij}$ and reward $w_{ij}r_i$ in a new graph $\mathcal{G}'$. Each vertex has its own local policy $\phi_{ij}(w_{ij}|s)$. Note that in the sense of mean-field approximation, we focus on neighbors and find a MRF: each $w_{ij}$ needs and only needs to be determined considering other $\{w_{ik}|k \in \mathcal{N}_i\setminus\{j\}\}$, because their actions are subject to the constraint $\sum_{j \in \mathcal{N}_i} w_{ij} = 1$. It accords with the adjacency relationship in $\mathcal{G}'$. According to Lemma A.1, it is also a Gibbs random field.

Now we consider the cliques that we factor $\boldsymbol{\phi}(\boldsymbol{w}|s)$ over. For $\forall i \in \mathcal{V}$, $\{d_{ij}|j \in \mathcal{N}_i\}$ should form a complete subgraph in $\mathcal{G}'$. Note that $d_{ij} \in \mathcal{G}'$ connects to $\{d_{ik}|k \in \mathcal{N}_i\setminus\{j\}\}$ and $\{d_{kj}|k \in \mathcal{N}_j\setminus\{i\}\}$, but only the former will form the maximal clique. Therefore, we have $\boldsymbol{\phi}(\boldsymbol{w}|s) \approx \prod_{i \in \mathcal{V}} \phi_i(w_i^{\text{out}}|s)$. Note that technically each agent $i$ can determine $\{w_{ij}|j \in \mathcal{N}_i\}$ simultaneously. We allow agent $i$ to take charge of $\phi_i(w_i^{\text{out}}|s)$ as its high-level policy which is a joint policy of the complete subgraph in $\mathcal{G}'$, so that we can turn the view back to $\mathcal{G}$ from $\mathcal{G}'$ and verify each agent's independence in the high level.

Besides, from Proposition 4.1, we approximately have: $q_i^{\phi_i}(s, w_i^{\text{out}}; \pi_{\mathcal{N}_i}) = v_i^{\pi_i}(s; w_i^{\text{in}}, \phi_{\mathcal{N}_i})$, where $q_i^{\phi_i}$ is the action-value function of $\phi_i$ given $\pi_{\mathcal{N}_i}$, $v_i^{\pi_i}$ is the value function of $\pi_i$ given $\phi_{\mathcal{N}_i}$ and conditioned on $w_i^{\text{in}}$. Let $Q_{\mathcal{N}_i}^{\phi}(s, \boldsymbol{w}; \boldsymbol{\pi}) \doteq \sum_{j \in \mathcal{N}_i} v_j^{\boldsymbol{\pi}}(s; w_j^{\text{in}}, \boldsymbol{\phi})$. Note that $\phi_i$ optimizes $Q_{\mathcal{N}_i}^{\phi}(s, \boldsymbol{w}; \boldsymbol{\pi})$, because only elements in $\{v_j^{\boldsymbol{\pi}}(s; w_j^{\text{in}}, \boldsymbol{\phi})|j \in \mathcal{N}_i\}$ correlate with $w_i^{\text{out}}$, while those in $\{v_j^{\boldsymbol{\pi}}(s; w_j^{\text{in}}, \boldsymbol{\phi})|j \in \mathcal{V}\setminus\mathcal{N}_i\}$ do not. After taking this uncorrelated set into account, we have an equivalent optimization of $Q_{\mathcal{V}}^{\phi}(s, \boldsymbol{w}; \boldsymbol{\pi})$, *i.e.*, the global objective. Therefore, each decentralized high-level policy shares the same optimization objective as the global one, and we can factorize $J_{\boldsymbol{\phi}}$ into $\{J_{\phi_i}|i \in \mathcal{V}\}$. $\square$

This proposition gives a factorization which is different from existing studies. First, our factorization differs from $\pi(\boldsymbol{a}|s) = \Pi_{i=1}^{N} \pi(a_i|s)$ (Zhang et al., 2018), since each agent needs to make decisions considering other agents' plan. Also in contrast to Qu et al. (2020a), they parameterize intention propagation by GNN and other neural networks to factorize the joint policy thoroughly, while we accept incomplete factorization and group indecomposable cliques by each agent to form high-level policies that are also decentralized and independent of each other.

## B   Algorithm

We describe LToS as Algorithm 1.

## C   Discussions on Training LToS

As a hierarchically decentralized MARL framework, LToS brings some challenges for training.

Selfishness Initializer. On the basis of a straightforward idea that one should generally focus more on its own reward than that of others when optimizing its own policy, the initial output of each high-level policy network is supposed to be higher on the sharing weight of its own than others. We choose to predetermine the initial selfishness to learn the high-level policy effectively. However, with normal initializers, the output of the high-level policy network will be evenly distributed initially. Therefore, we use a special *selfishness initializer* for each high-level policy network instead. As we use the softmax to produce the weights, which guarantees the constraint: $\sum_{j \in \mathcal{N}_i} w_{ij} = 1, \forall i \in \mathcal{V}$, we specially set the bias of the last fully-connected layer so that each decentralized high-level policy network tends to keep for itself the same reward proportion as the given selfishness initially. The rest of reward is still evenly distributed among neighbors. LToS learns started from such initial weights, while *fixed* LToS uses such weights throughout each experiment. Moreover, we use grid search to find the best selfishness for *fixed* LToS in *traffic* and *jungle*. For *prisoner* we deliberately set the selfishness to $0.5$ so that *fixed* LToS directly optimizes the average return.

**Algorithm 1** LToS

1: Initialize $\phi_i$ parameterized by $\theta_i$ and $\pi_i$ parameterized by $\mu_i$ for each agent $i$ ($\phi_i$ is learned by DDPG and $\pi_i$ is learned by DGN, and they share the Q-network)
2: **for** $t = 0$ to $T$ **do**
3:     **for** each agent $i$ **do**
4:         exchange observations and get $o_i$
5:         $w_i^{\text{out}} \leftarrow \phi_i(o_i)$ with exploration
6:         exchange $w_i^{\text{out}}$ and get $w_i^{\text{in}}$
7:         $a_i \leftarrow \pi_i(o_i; w_i^{\text{in}})$ with exploration
8:         execute $a_i$, obtain $r_i$, and transition to $o_i' = o_{i,t+1}$
9:         exchange $r_i$ and get $r_i^w$
10:       store $(o_i, w_i^{\text{in}}, a_i, r_i^w, o_i', \mathcal{N}_i)$ in $\mathcal{B}_i$
11:     **end for**
12:     **if** $t \bmod update\_frequency = 0$ **then**
13:         **for** each agent $i$ **do**
14:             sample a minibatch from replay buffer $\mathcal{B}_i$: $D = \{(o_i, w_i^{\text{in}}, a_i, r_i^{\boldsymbol{w}}, o_i', \mathcal{N}_i)\}$
15:             exchange $w_i^{\text{out}'} \leftarrow \phi_i'(o_i')$ and get $w_i^{\text{in}'}$
16:             set $y_i \leftarrow r_i^{\boldsymbol{w}} + \gamma q_i^{\pi_i}(o_i', a_i'; w_i^{\text{in}'})|_{a_i' = \pi_i'(o_i'; w_i^{\text{in}'})}$
17:             update $\mu_i$ by $\nabla_{\mu_i} \mathbb{E}_D(y_i - q_i^{\pi_i}(o_i, a_i; w_i^{\text{in}}))^2$
18:             exchange $w_i^{\text{out}} \leftarrow \phi_i(o_i)$ and get $w_i^{\text{in}}$
19:             compute $g_i^{\text{in}} = \nabla_{w_i^{\text{in}}} q_i^{\pi_i}(o_i, \arg\max_{a_i} q_i^{\pi_i}; w_i^{\text{in}})$
20:             exchange $g_i^{\text{in}}$ and get gradient $g_i^{\text{out}}$ for $w_i^{\text{out}}$
21:             update $\theta_i$ by $\frac{1}{|D|} \sum_{o_i \in D} (\nabla_{\theta_i} \phi_i(o_i))^{\mathsf{T}} g_i^{\text{out}}$
22:             softly update target networkrs $\theta_i'$ and $\mu_i'$
23:         **end for**
24:     **end if**
25: **end for**

**Unified Pseudo Random Number Generator.** LToS is learned in a decentralized manner. This incurs some difficulty for experience replay. As each agent $i$ needs $w_i^{\text{in}}$ to update network weights for both high-level and low-level policies, it should sample from its buffer a batch of experiences where each sampled experience should be synchronized across the batches of all agents (*i.e.*, the experiences should be collected at a same timestep). To handle this, all agents can simply use a unified pseudo random number generator and the same random seed.

**Different Time Scales.** As many hierarchical RL methods do, we can set the high-level policy to running at a slower time scale than the low-level one. Proposition 1 still holds if we expand $v_i^{\boldsymbol{\pi}}$ for more than one step forward. Assuming the high-level policy runs every $M$ timesteps, we can fix $w_i^{\text{out}} = w^{\text{out},t+1} = \cdots = w^{\text{out},t+M-1}$. $M$ is referred to as *action interval* in Table 5.

**Infrequent Parameter Update with Small Learning Rate.** Based on the continuity of $\boldsymbol{w}$, a small modification of $\phi$ means a slight modification of local reward functions, and will intuitively result in an equally slight modification of the low-level value functions. This guarantees the low-level policies are highly reusable.

**Unordered Output.** Essentially, the output of high-level policy network is unordered and has a one-to-one match with each neighbor as input. The output of deep neural network, however, is generally ordered and has trouble in varying with the input order. To settle this, we take advantage of DGN which is insensitive of neighbor order as input. Besides, we modify the structure to make the output keep consistency with the neighbor part of input in the relative order.

## D   Hyperparamaters

As three experimental scenarios are quite different, we may use different hyperparameters. Note that we also tuned the hyparameters for the baselines with grid search. Table 4 summarizes the hyperparameters of DQN, DGN that also serves as the low-level network of LToS, and IP. We choose

the setting of original DGN in *jungle* while the setting of Wei et al. (2019) in *traffic* for consistency. Table 5 summarizes the hyperparameters of the high-level network of LToS, which are different from the low-level network. Table 6 summarizes the hyperparameters of NeurComm and ConseNet, which adhere to the implementation (Chu et al., 2020). In addition, for tabular Coco-Q, the step-size parameter is $0.5$, and for IP, the regularizer factor is $0.2$. We adopt soft update for both high-level and low-level networks and use an Ornstein-Uhlenbeck Process (abbreviated as OU) for high-level exploration.

Both *fixed* LToS and NeurComm exploit static reward shaping, but they adopt different reward shaping schemes which are hard to compare directly. We consider a simple indicator: Self Neighbor Ratio (SNR), the ratio of reward proportion that an agent chooses to keep for itself to that it obtains from a single neighbor. As the rest reward is evenly shared with neighbors in LToS, for each agent $i$, we have $\text{SNR} = \text{selfishness}/\text{1-selfishness} \times (|\mathcal{N}_i| - 1)$ for LToS, and $\text{SNR} = 1/\alpha$ for NeurComm where $\alpha$ is the spatial discount factor. We adjust the initial selfishness and $\alpha$ to set the SNR of both methods at the same level for fair comparison.

Table 4: Hyperparameters for DQN, DGN (also serves as the low-level policy network of LToS), and IP.

| Hyperparamater | *Prisoner* | *Jungle* | *Traffic*-$6 \times 6$ | *Traffic-Shenzhen* |
|---|---|---|---|---|
| sample size | 10 | 10 | 1,000 | 1,000 |
| batch size | 10 | 10 | 20 | 20 |
| buffer capacity | 200,000 | 200,000 | 10,000 | 10,000 |
| $\epsilon_{start}, \epsilon_{decay}, \epsilon_{end}$ | 0.8/1/0.8 | 0.6/0.996/0.01 | 0.4/0.9/0.05 | 0.4/0.9/0.05 |
| initializer | random normal | random normal | random normal | random normal |
| optimizer | Adam | Adam | RMSProp | RMSProp |
| learning rate | 1e-3 | 1e-4 | 1e-3 | 1e-3 |
| $\gamma$ | 0.99 | 0.96 | 0.8 | 0.8 |
| $\tau$ for soft update | 0.1 | 0.01 | 0.1 | 0.1 |
| # MLP units | 32 & 32 | 512 & 128 | 32 & 32 | 32 & 32 |
| MLP activation | ReLU | ReLU | ReLU | ReLU |
| # encoder MLP layers | 2 | 2 | 2 | 2 |
| # attention heads for DGN | 4 | 4 | 1 | 1 |

Table 5: Hyperparameters for the high-level policy network of LToS

| Hyperparamater | *Prisoner* | *Jungle* | *Traffic*-$6 \times 6$ | *Traffic-Shenzhen* |
|---|---|---|---|---|
| update frequency | 1 step | 100 episodes | 20 episodes | 50 episodes |
| action interval | 1 step | 1 step | 15 steps | 15 steps |
| sample size | 2,000 | 5,000 | 1,000 | 1,000 |
| batch size | 32 | 32 | 20 | 20 |
| noise for exploration | $\epsilon$ + Gaussian | OU | OU | OU |
| noise parameter | $\epsilon = 0.8, \sigma = 1$ | $\sigma = 0.025\epsilon$ | $\sigma = 0.25\epsilon$ | $\sigma = 0.25\epsilon$ |
| initializer | selfishness | selfishness | selfishness | selfishness |
| initial selfishness | 0.5 | 0.5 | 0.8 | 0.9 |
| optimizer | SGD | SGD | SGD | SGD |
| learning rate | 1e-1 | 1e-4 | 1e-3 | 1e-3 |
| last MLP layer activation | softmax | softmax | softmax | softmax |

Table 6: Hyperparameters for NeurComm, ConseNet, LIO and QMIX

| Hyperparamater | *Prisoner* | *Jungle* | *Traffic*-$6 \times 6$ | *Traffic-Shenzhen* |
|---|---|---|---|---|
| initializer | orthogonal | orthogonal | orthogonal | orthogonal |
| optimizer | RMSProp | RMSProp | RMSProp | RMSProp |
| learning rate | 5e-3 | 5e-5 | 5e-4 | 5e-4 |
| # MLP units | 20 | 512 & 128 | 16 | 16 |
| MLP activation | ReLU | ReLU | ReLU | ReLU |
| # cell state units | 20 | 512 | 16 | 16 |
| # hidden state units | 20 | 512 | 16 | 16 |
| RNN type for NeurComm and ConseNet | LSTM | LSTM | LSTM | LSTM |
| RNN type for QMIX | GRU | GRU | GRU | GRU |
| hypernetwork layer1 units for QMIX | $2 \times 20$ | $20 \times 512$ | $36 \times 16$ | $36 \times 16$ |
| hypernetwork layer2 units for QMIX | 20 | 512 | 16 | 16 |
| $\alpha$ for NeurComm | 1 | 0.33 | 0.1 | 0.1 |
| $\epsilon_{start}, \epsilon_{decay}, \epsilon_{end}$ for LIO | 0.8/0.99/0.01 | 0.6/0.996/0.01 | 0.2/0.9/0.01 | 0.2/0.9/0.01 |
| $\alpha_{\theta}$ for LIO | 1 | 1e-4 | 1e-4 | 1e-4 |
| $R_{max}$ for LIO | 2 | 3 | 0.1 | 0.1 |