# OpenReview forum: "Learning to Share in Networked Multi-Agent Reinforcement Learning"
_NeurIPS.cc/2022/Conference — NeurIPS 2022 Accept_

### Official Review · Reviewer_cVVf · 2022-07-10

**Rating:** 6
**Confidence:** 3
**Soundness:** 3 good
**Presentation:** 4 excellent
**Contribution:** 3 good

**Summary:**

This paper considers a networked multi-agent reinforcement learning (MARL) setting, where the goal is to maximize the sum of discounted cumulative rewards of agents, where at each step, each agent's reward depends on its own action, as well as that of its neighbors in the network graph. A hierarchical scheme is proposed, where the high-level policy learns what fraction of each agent's reward to be shared with its neighbors, while the low-level policy is trained to maximize the agent's local objective induced by the high-level policy. Numerical results on three different multi-agent environments demonstrate the efficacy of the proposed method when compared to multiple state-of-the-art MARL baselines in the literature.

**Questions:**

- I believe a line of work on using graph neural networks in MARL is missing from the Related Work Section, including, but not the following:

  - Liu, Yong, Weixun Wang, Yujing Hu, Jianye Hao, Xingguo Chen, and Yang Gao. "Multi-agent game abstraction via graph attention neural network." In Proceedings of the AAAI Conference on Artificial Intelligence, vol. 34, no. 05, pp. 7211-7218. 2020.
  - Liu, Iou-Jen, Raymond A. Yeh, and Alexander G. Schwing. "PIC: permutation invariant critic for multi-agent deep reinforcement learning." In Conference on Robot Learning, pp. 590-602. PMLR, 2020.
  - Naderializadeh, Navid, Fan H. Hung, Sean Soleyman, and Deepak Khosla. "Graph convolutional value decomposition in multi-agent reinforcement learning." arXiv preprint arXiv:2010.04740 (2020).
  - Shen, Siqi, Yongquan Fu, Huayou Su, Hengyue Pan, Peng Qiao, Yong Dou, and Cheng Wang. "Graphcomm: A graph neural network based method for multi-agent reinforcement learning." In ICASSP 2021-2021 IEEE International Conference on Acoustics, Speech and Signal Processing (ICASSP), pp. 3510-3514. IEEE, 2021.
  - Li, Sheng, Jayesh K. Gupta, Peter Morales, Ross Allen, and Mykel J. Kochenderfer. "Deep Implicit Coordination Graphs for Multi-agent Reinforcement Learning." In Proceedings of the 20th International Conference on Autonomous Agents and MultiAgent Systems, pp. 764-772. 2021.
  - Siu, Chapman, Jason Traish, and Richard Yi Da Xu. "Dynamic Coordination Graph for Cooperative Multi-Agent Reinforcement Learning." In Asian Conference on Machine Learning, pp. 438-453. PMLR, 2021.

- Does the last term in the second equation in (3) need to be replaced by $v_i^{\boldsymbol{\pi}}(s';\boldsymbol{w},\phi)$?

- It is slightly unclear whether, at the end of the day, the average **shaped** reward is being optimized over the entire network or the average **original** reward.

- Please define the *selfishness* metric precisely using the notion already introduced in the paper.

- Following up on the above comment, the selfishness metric is an aggregate metric, which does not highlight more details on the learned reward sharing mechanism. Are there any deeper insights on the learned high-level policies that the authors can share in terms of what the weights $\\{w_{ij}\\}_{ij \in \mathcal{E}}$ depend on, for example, are they correlated with how much agents are geographically close, etc.?

- Could you please add more details on how the outer bounds for the *jungle* and *traffic* environments (i.e., upper and lower bounds, respectively) were derived?

**Limitations:**

The authors describe some of the limitations of implementing the proposed method, but as I mentioned above, I would also love to see a discussion of how the graph construction impacts the performance of the proposed method. Also, as MARL, in general, might have potential negative societal impacts, it would be helpful to also add those as potential application domains where LToS might enable such negative impacts.

**Strengths And Weaknesses:**

The paper is clearly written and easy to read. The considered problem is timely and interesting, and the proposed solution shows promising performance in practice. In my opinion, the considered number of environments and baselines to compare with in the experiments is a great positive point of the paper.

That being said, the main weakness of the paper I can think of is the construction of the network graph. It seems that in the experiments, a nearest-neighbor type of graph is used, where each agent is connected to a few neighboring agents in its physical vicinity. However, such a geometric and symmetric graph construction method might be suboptimal in practice. In general, I wonder how the graph should be constructed and whether there is an automatic way to do so. Some ablation studies on the impact of graph density (for example, controlled by the number of neighbors per node) might be helpful to shed some light on this.

---

> ### Author Response · Authors · 2022-08-02
> **To Reviewer cVVf:**
>
> > the construction of the network graph
>
> Good question. In networked MARL, the graph is typically assumed to be given or simply structured by vicinity [1,2,3], since the graph structure can be too complex [4]. As for the study on graph density, we have added new experiments in *jungle*,  we choose the number of neighbors to be 1, 2, 3, and 4. As illustrated Figure 8 in the revision of Appendix,  the number of neighbors indeed affects the performance. By now we choose to consider the number of neighbors as a hyperparameter to tune as [2] do, and #neighbors=3 is the best in *jungle*.
>
> [1] Value propagation for decentralized networked deep multi-agent reinforcement learning, C. Qu et al., NeurIPS 2019.
> [2] Intention propagation for multi-agent reinforcement learning, Qu et. al., 2020.
> [3] Scalable multi-agent reinforcement learning for networked systems with average reward, G. Qu et al., NeurIPS 2020.
> [4] Self-Organized Polynomial-Time Coordination Graphs, Yang et al., ICML 2022.
>
> > a line of work on using graph neural networks in MARL is missing
>
> Thanks for bringing up these papers. We will expand the section of related work to include these works. Besides, we consider our LToS more of a new hierarchical MARL framework than a new method, so it's not restricted to DDPG+DGN but can be realized by diverse combinations of methods. We believe LToS can achieve better performance when implemented by these newer approaches.
>
> > Does the last term in the second equation in (3) need to be replaced by $v_i^{\boldsymbol{\pi}}(s';\boldsymbol{w},\boldsymbol{\phi})$?
>
> We are afraid not. Here the first line is Bellman equation of the bootstrapped V-function $v_i^\pi (s; \boldsymbol{\phi})$, and the second line uses V-function to define $v_i^\pi (s; \boldsymbol{w}, \boldsymbol{\phi})$ conditioned on $\boldsymbol{w}$.
>
> > It is slightly unclear whether, at the end of the day, the average shaped reward is being optimized over the entire network or the average original reward.
>
> The average original reward is optimized over the entire network, by the optimization of shaped local reward from the perspective of each agent. Note that the sum of shaped local reward of all agents is equal to the sum of original rewards of all agents.
>
> > about selfishness
>
> Sorry that we are restricted to the page limit. **The proportion of the reward an agent keeps for itself is called *selfishness*.** For details, please see {Selfishness Initializer} and {Hyperparameter} part of our appendix in the supplementary material.
>
> > what the weights $\{w_{ij}\}_{ij \in \mathcal{E}}$ depend on?
>
> Good question. This is very complicated. We would say that the weights should depend on the task, its inherent MDP, and the converged policy, not simply how much two agents are geographically close.
>
> >  the outer bounds for the jungle and traffic environments
>
> In *jungle*, we give the bound by solving the maximum weight matching problem in a bipartite graph (foods and agents). In *traffic*, we give the bound by optimal traffic dispatch. Note that they both require global information and assignment, so the two bounds are loose and unrealistic.
>
> > about negative societal impacts
>
> Thanks for your advice and we will add a discussion on potential negative societal impacts. Meanwhile, we believe our framework will have more positive societal impacts than negative ones. For example, by following the setting of networked MARL, information of non-neighboring nodes can be preserved, and thus their privacy & safety can be protected.

---

> > ### Comment · Reviewer_cVVf · 2022-08-04
> > **Thank you!**
> >
> > Thank you very much for your response. I will keep my score unchanged.

---

> > > ### Author Response · Authors · 2022-08-08
> > > **Thanks**
> > >
> > > We are glad that all the comments of the reviewer have been addressed. Thanks to the reviewer for acknowledging the paper's contribution.

---

### Official Review · Reviewer_gcHc · 2022-07-11

**Rating:** 6
**Confidence:** 3
**Soundness:** 3 good
**Presentation:** 3 good
**Contribution:** 3 good

**Summary:**

This paper considers networked cooperative MARL settings where agents can communicate with a set of neighboring agents. The authors propose an approach for sharing rewards among neighboring agents in which a policy for sharing rewards between each neighboring node is learned along with the main policy learning task within a bilevel alternating optimization framework. A series of multi-agent experiments are conducted while learning the main policy leveraging a DQN with a reward sharing policy based on DDPG. The proposed LToS framework outperforms a set of MARL baselines on the prisoner, jungle, and traffic domains considered for evaluation. Additionally, LToS is demonstrated to learn a more effective conditional reward sharing policy than is learned by a fixed reward sharing policy baseline.

**Questions:**

As I am very interested in the comparison with the fixed or average reward sharing approach, I had some clarifying questions to ask about the "fixed LToS" baseline:
- For “fixed LToS” what are the sharing weights fixed at? Average over all neighbors?
- Performance improvements on the “prisoner” domain are not terribly clear. Why does fixed LTOS perform so poorly relative to the DQN baseline? I would have expected the same or better performance because of reward sharing, why would it be so much worse?
- Can you explain this comment about fixed LTOS? I am having a hard time understanding what is meant by this seemingly crucial point: “This is because fixed reward sharing does not adapt to the dynamic topology. By learning proper reward sharing and adjusting to changing circumstances, LToS outperforms all other baselines.”

**Limitations:**

It would be nice to expand a bit more in the main text about the significance of proposition 4.2 particularly with reference to the Deep MARL based experiments conducted.

**Strengths And Weaknesses:**


Strengths
- The overall networked communication setting of this paper is an interesting one, and learning a reward sharing policy seems like a potentially important operation in this setting. I am not aware of any other paper that explicitly learned to share rewards.
- Proposition 4.2 seems to be a potentially interesting result.
- Their are a nice set of experiments across a variety of multi-agent domains that illustrate the performance of the proposed framework.

Weaknesses
- The paper does not compare empirically or theoretically with other frameworks that learn to share in cooperative MARL such as action advising [1], goal-state advising [2], or sharing abstract messages [3, 4, 5]. It seems like reward sharing is only one aspect of what could potentially be useful to learn to share along with states, actions, and more abstract forms of communication.
- The motivation for learning dynamic weights for sharing rewards over time was not made clearly and missing details about how the fixed weight sharing was implemented made it hard to know what conclusions to draw from the experiments. Given the global objective, why not just send the average local reward to each agent? Can you explain more about how dynamic weight sharing schemes work and why they would be preferable to the best fixed weight reward sharing approach?
- Minor Comment: Figure 3 is very small.

Overall, I am on the fence about this paper and look forward to receiving a response form the authors to the questions I have raised. I lean slightly towards rejection at the moment given that I am a bit unclear about the motivation for the approach and have questions about the baselines in the experiments.

[1] "Learning to Teach in Cooperative Multiagent Reinforcement Learning" Omidshafiei et al., AAAI-2019.
[2] "Learning Hierarchical Teaching Policies for Cooperative Agents" Kim et al., AAMAS-2020.
[3] "Emergence of Grounded Compositional Language in Multi-Agent Populations" Mordatch and Abbeel, AAAI-2018.
[4] "Biases for Emergent Communication in Multi-agent Reinforcement Learning" Eccles et al., NeurIPS-2019.
[5] "TarMAC: Targeted Multi-Agent Communication" Das et al., ICML-2019.

---

> ### Author Response · Authors · 2022-08-02
> **To Reviewer gcHc:**
>
> > About related work
>
> Thanks for bringing up these related works and we will expand the section of related work to include these works. [1] investigated learning to coordinate and teach, but limited to two-agent cases, at least from the experiments. However, how to extend to a large number of agents is unclear. [2] is a method of centralized training with decentralized execution for global reward setting. For this kind of methods, we have included QMIX as a baseline. Please refer to the response to reviewer 98u6 for why this kind of methods does not work well in networked MARL setting. [3] and [4] focus on discrete communication (symbols) between agents and aim at explaining the emergence of communication. [5] uses attention to weigh the incoming messages, assuming each agent can receive information from all other agents. This contradicts the networked MARL setting. For the line of research on abstract forms of communication, we have compared with NeurComm [6] and intention propagation [7], which are tailored for networked MARL setting.
>
>
> [1] Learning to Teach in Cooperative Multiagent Reinforcement Learning, Omidshafiei et al., AAAI2019.
>
> [2] Learning Hierarchical Teaching Policies for Cooperative Agents, Kim et al., AAMAS2020.
>
> [3] Emergence of Grounded Compositional Language in Multi-Agent Populations, Mordatch and Abbeel, AAAI2018.
>
> [4] Biases for Emergent Communication in Multi-agent Reinforcement Learning, Eccles et al., NeurIPS2019.
>
> [5] TarMAC: Targeted Multi-Agent Communication, Das et al., ICML2019.
>
> [6] Multi-agent reinforcement learning for networked system control, Chu et al., ICLR2020.
>
> [7] Intention propagation for multi-agent reinforcement learning, Qu et. al., 2020.
>
>
> > About fixed weight sharing and motivation for learning dynamic weights
>
> For fixed weight sharing, each agent keeps a proportion of reward for itself (called selfishness) and the rest is averaged over all neighbors. The selfishness of fixed LToS is determined by grid search in jungle and traffic. Please see the section of Selfishness Initializer and Hyperparameter in the supplementary material for details. The fixed-weight reward sharing cannot work well, because the environment and neighbors of an agent are dynamic and hence it is preferable to dynamically adjust these weights. To verify this, we can compare LToS and fixed LToS, both of which as the same initial weights. In jungle, the topology is dynamic. If the weights are fixed, the agent cannot abandon old and construct new cooperative relationship with new neighbors. As for traffic, the weight visualization (Figure 6 ) shows that handling heavier traffic requires more diverse reward-sharing weights to promote more sophisticated cooperation.  As for sending the average local reward to each agent, it is similar to global reward setting like QMIX. However, we have shown QMIX does not work well in the experiments.
>
> > About the performance of fixed LToS in prisoner.
>
> Sorry for the unclear Figure 3. The dark blue curve (better than fixed LToS) in Figure 3a is Coco-Q (it has a similar color to DQN), not DQN. Actually, the curves of DQN and DGN are always around reward=0.5 and covered by other curves. So, fixed LToS actually outperforms DQN. We have addressed this in the revision.

---

> > ### Comment · Reviewer_gcHc · 2022-08-09
> > **Re: To Reviewer gcHc:**
> >
> > Thank you for providing clarification regarding the fixed weight sharing baseline and the need for dynamic weight sharing. That makes sense to me. I also appreciate the author’s clarification about Figure 3 and connections to some of the related work I mentioned. As some of my main concerns about the paper have been addressed, I have increased my score accordingly.

---

> ### Author Response · Authors · 2022-08-08
> **Looking forward to your feedback**
>
> We first would like to thank the reviewer's efforts and time in reviewing our work. As the author-reviewer discussion is ending soon, we were wondering if our responses have resolved your concerns. We will be happy to have further discussions with the reviewer if there are still remaining questions.

---

### Official Review · Reviewer_98u6 · 2022-07-11

**Rating:** 5
**Confidence:** 4
**Soundness:** 2 fair
**Presentation:** 3 good
**Contribution:** 2 fair

**Summary:**

This paper proposes a hierarchical method for networked multi-agent reinforcement learning, where the agents can share their local rewards with nearby agents over a partially connected network. In the proposed method, each agent optimized its value function in a decentralized manner, using a weighted sum of local rewards shared by its neighbors. To maximize the global objective, it uses a two-level structure of policies: A high-level policy learns optimal weights for shared rewards, and a low-level policy learns to optimize the local objective induced by the weighted sum of shared rewards. Propositions in this paper imply that optimizing individual objectives can achieve optimizing the global objective of the global perspective, and the experimental results demonstrate the effectiveness of the proposed method in three environments.

**Questions:**

1. Is it fair to compare the centralized training algorithm with the setting suitable for decentralized training? Even if it is, why does QMIX have low performance in the prisoner environment?

2. The authors chose DGN to optimize the low-level policy with shaped rewards. Is there a specific reason for using DGN instead of other algorithms? It would be meaningful to make a novel framework specialized in optimizing the shaped rewards.

**Limitations:**

1. Networked MARL setting seems to be a subfield of MARL that has recently become popular. In this setting, rewards are given globally or often individually. But I'm not sure how useful the individual reward setting is. The global reward setting is more general and actively studied regardless of centralized or decentralized training. I think the individual reward setting presented in this paper is quite narrow.

2. The proposed method is limited only to applying to environments in which each agent has a fixed number of neighbors. For example, in the jungle environment, each agent has 3 closest agents as neighbors, and the proposed method can be applied to such environments. On the other hand, we can consider a general setting of choosing neighbors, that is, defining neighbors as all agents within a d ball from the agent. Such a definition makes the number of neighbors change over time, and it would be meaningful if the high policies adapt to the time-varying cases.

**Strengths And Weaknesses:**

Strength:
1. The hierarchical structure for optimizing the global objective through reward sharing is well organized and may be applied in various scenarios with massive agents in the real world.
2. Propositions show well that two-level policies trained in a decentralized way can reach global objectives.
3. Evaluations explain and visualize the effectiveness of the proposed method well.

Weaknesses:
1. The main idea of this paper, which maximizes the weighted sum of local rewards from its neighbors to enhance cooperative behavior, seems a little lacking in novelty.
2. All proofs assume a global perspective and the process of bringing this to the decentralized setting is explained poorly.

---

> ### Author Response · Authors · 2022-08-02
> **To Reviewer 98u6:**
>
> > About novelty
>
> Maximizing the weighted sum of local rewards is straightforward. However, **the novelty is how to determine the weights, i.e., how to share rewards**. As verified empirically by ablation study, a fixed weight is inadequate to promote cooperation among agents.
>
> > About proof
>
> We choose the global perspective for the sake of convenience since the optimization objective is a joint target for all agents. The transition to decentralized setting was indeed described by Proposition 2 and lines 148-163. We believe it is explained thoroughly.
>
> > About why compare with the centralized training algorithm, QMIX
>
> First, as you mentioned in the review, the global reward setting is more general, so **a natural question is whether the method for global reward setting can solve individual reward setting. QMIX serves for this purpose,** since QMIX can be easily set to optimize the sum of individual rewards.
>
> **Why does not QMIX perform well?**  This is not a surprise, as it is observed by existing study [1]. One major reason is that QMIX is agnostic to the fact that the global reward is a sum of individual rewards and it has to learn that. However, this can be hardly matched by the learned implicit credit assignment of QMIX. In prisoner, We would say QMIX does not perform that bad (close to the best performance), compared its performance in jungle and traffic. This is because in prisoner there are only two agents, QMIX eventually learns the correct credit assignment. However, this is not the case in the task with more agents as in jungle and traffic.
>
> [1] Intention propagation for multi-agent reinforcement learning, Qu et. al., 2020.
>
>
> > About the reason to choose DGN
>
> We consider our LToS more of a new hierarchical MARL framework than a new method, so it is not restricted to DDPG+DGN but can be realized by diverse combinations of methods. We currently employed DGN in our experimental because it is capable to handle communication while others (DQN, DDPG, A2C) are not, and it has shown its advantage over others like CommNet (Sukhbaatar et al. NeurIPS 2016).
>
> > Individual reward setting is quite narrow
>
> **We could not agree with this.** There are many real applications that are individual reward settings. To name a few, in wireless networks, each base station or access point has its own reward, for example, power consumption, how to cooperatively optimize total power consumption. In inventory management, there are many warehouses and each has own reward, for example, profit. How to optimize the total profit.
>
> Certainly, it is not as general as global reward setting, but the method for global reward setting cannot well solve individual reward setting as mentioned above.
>
>
> > The proposed method is limited only to applying to environments in which each agent has a fixed number of neighbors.
>
> In jungle, we follow the setting of the DGN paper. Due to the TensorFlow implementation of DGN (please check the code in Supplementary), it currently cannot handle dynamic number of neighbors. But our method is limited to that.

---

> > ### Comment · Reviewer_98u6 · 2022-08-04
> > **Response to the authors**
> >
> > I thank the authors for their response. I have read the response, and most of the questions I raised, are solved. However, I still have some remaining questions as follows.
> >
> > Q1. Is it fair to compare algorithms using only global rewards including QMIX with frameworks using individual rewards? It is a critical disadvantage that the method cannot use individual rewards in an environment where those are given.
> >
> > Q2. In the Jungle environment, the increase in the number of neighbors seems to have a negative impact on performance. What is the reason for this performance degradation? Does the performance decrease as the number of neighbors increases?

---

> > > ### Author Response · Authors · 2022-08-06
> > > **Response to reviewer 98u6**
> > >
> > > It is great that we have addressed most of your questions.
> > >
> > > > Q1. Is it fair to compare algorithms using only global rewards including QMIX with frameworks using individual rewards? It is a critical disadvantage that the method cannot use individual rewards in an environment where those are given.
> > >
> > > To be honest, it is *unfair*. But, the purpose of comparing with QMIX is actually for broader audiences, and this shows that networked MARL settings cannot be easily solved by the general method using only global rewards. This was a question we were asked before.
> > >
> > > > Q2. In the Jungle environment, the increase in the number of neighbors seems to have a negative impact on performance. What is the reason for this performance degradation? Does the performance decrease as the number of neighbors increases?
> > >
> > > This is an interesting point. To further investigate this, we carried out an additional experiment on #neighbor=5. The following table shows the results for different numbers of neighbors, where with the increase of #neighbor, the performance increases first and then decreases in an indefinitive way. We think the reason may be that when the graph density increases, the learning difficulty also increases as the agent needs to decide how to share rewards with more neighbors. Beyond a certain threshold, the learning difficulty outweighs the gain.
> > > This is similar to learning to communicate, where the graph density increases, the gain of communication decreases due to learning difficulty [1].
> > >
> > > However, as in networked MARL the graph structure is typically assumed to be sparse, and given, this should not be a problem.
> > >
> > > [1] Learning Individually Inferred Communication for Multi-Agent Cooperation, NeurIPS 2020.
> > >
> > > | # neighbors  | LToS  |
> > > |  :----: | :----:  |
> > > |	1 |	0.83 |
> > > | 	2 |	0.84 |
> > > |	3 |	**0.86** |
> > > |	4 |	0.66 |
> > > |	5 |	0.80 |
> > >
> > > Table 3. LToS on different number of neighbors in *jungle*. The new results have been added to Appendix.

---

> > > > ### Comment · Reviewer_98u6 · 2022-08-08
> > > > **Respond to the authors**
> > > >
> > > > Thanks again for your reply. All my concerns about this paper are solved, so I am increasing my score to 5.

---

> > > > > ### Author Response · Authors · 2022-08-08
> > > > > **Thanks for raising the score!**
> > > > >
> > > > > We are glad that all the concerns of the reviewer are addressed. We thank the reviewer for raising the score!

---

### Meta-Review · Area_Chair_Bnaq · 2022-08-26

**Recommendation:** Accept
**Confidence:** Less certain

**Metareview:**

While the scores are borderline, reviewers found the paper interesting and the experiments convincing, so I recommend acceptance. While there were originally concerns about the appropriateness of comparison with QMIX and the relationship to related work, I think these were adequately addressed in the rebuttal.

**Award:**

No

---

### Decision · Program_Chairs · 2022-09-14

Accept